# Mexican *Ganoderma Lucidum* Extracts Decrease Lipogenesis Modulating Transcriptional Metabolic Networks and Gut Microbiota in C57BL/6 Mice Fed with a High-Cholesterol Diet

**DOI:** 10.3390/nu13010038

**Published:** 2020-12-24

**Authors:** Sandra L. Romero-Córdoba, Ivan Salido-Guadarrama, María E. Meneses, Giulia Cosentino, Marilena V. Iorio, Elda Tagliabue, Nimbe Torres, Mónica Sánchez-Tapia, Myrna Bonilla, Ivan Castillo, Beatriz Petlacalco, Armando R. Tovar, Daniel Martínez-Carrera

**Affiliations:** 1Biochemistry Department, Instituto Nacional de Ciencias Médicas y Nutrición Salvador Zubirán (INCMNSZ), Mexico City 14080, Mexico; sromero_cordoba@hotmail.com; 2Departamento de Medicina Genómica y Toxicología Ambiental, Instituto de Investigaciones Biomédicas, Universidad Nacional Autónoma de México, Mexico City 04510, Mexico; 3Molecular Targeting Unit, Fondazione IRCCS Istituto Nazionale dei Tumori, 20133 Milan, Italy; giulia.cosentino@istitutotumori.mi.it (G.C.); marilena.iorio@istitutotumori.mi.it (M.V.I.); elda.tagliabue@istitutotumori.mi.it (E.T.); 4Laboratorio de Biología Computacional, Instituto Nacional de Enfermedades Respiratorias Ismael Cosío Villegas (INER), Mexico City 14080, Mexico; silvervann@gmail.com; 5Centre of Biotechnology of Medicinal, Functional, and Edible Mushrooms, Colegio de Postgraduados (CP), Campus Puebla, Boulevard Forjadores de Puebla no. 205, Puebla 72760, Mexico; meneses.eugenia@colpos.mx (M.E.M.); myrna@colpos.mx (M.B.); castillo.ivan@colpos.mx (I.C.); petlacalco.beatriz@colpos.mx (B.P.); 6CONACYT-Colegio de Postgraduados (CP), Campus Puebla, Boulevard Forjadores de Puebla 205, Puebla 72760, Mexico; 7Departamento de Fisiología de la Nutrición, Instituto Nacional de Ciencias Médicas y Nutrición Salvador Zubirán (INCMNSZ), Mexico City 14080, Mexico; nimbester@gmail.com (N.T.); qfbmoniktc@gmail.com (M.S.-T.)

**Keywords:** hypercholesterolemia, *Ganoderma lucidum*, transcriptional profiles, lipid catabolism, microbiota

## Abstract

Prevention of hyperlipidemia and associated diseases is a health priority. Natural products, such as the medicinal mushroom *Ganoderma lucidum* (*Gl*), have demonstrated hypocholesterolemic, prebiotic and antidiabetic properties. However, the underlying transcriptomic mechanisms by which *Gl* exerts bioactivities are not completely understood. We report a comprehensive hepatic and renal transcriptome profiling of C57BL/6 mice under the consumption of a high-cholesterol diet and two standardized *Gl* extracts obtained from basidiocarps cultivated on conventional substrate (*Gl*-1) or substrate containing acetylsalicylic acid (ASA; *Gl*-2). We showed that *Gl* extracts modulate relevant metabolic pathways involving the restriction of lipid biosynthesis and the enrichment of lipid degradation and secretion. The *Gl*-2 extract exerts a major modulation over gene expression programs showing the highest similarity with simvastatin druggable-target-genes and these are enriched more in processes related to human obesity alterations in the liver. We further show a subset of *Gl*-modulated genes correlated with *Lactobacillus* enrichment and the reduction of circulating cholesterol-derived fats. Moreover, *Gl* extracts induce a significant decrease of macrophage lipid storage, which occurs concomitantly with the down-modulation of Fasn and Elovl6. Collectively, this evidence suggests a new link between *Gl* hypocholesterolemic and prebiotic activity, revealing thereby that standardized Mexican *Gl* extracts are a novel transcriptome modulator to prevent metabolic disorders associated with hypercholesterolemia.

## 1. Introduction

Cholesterol is a lipid essential for cell functions and the structure of cell membranes. However, epidemiological, metabolic and experimental studies have identified high cholesterol concentration as a major risk factor for metabolic and cardiovascular diseases [1]. Hypercholesterolemia is a condition showing an excessive amount of serum total cholesterol or low-density lipoprotein (LDL) [2]. Other major public health problems, such as coronary heart diseases, obesity and metabolic syndromes positively correlate with hypercholesterolemia [3]. Thus, the prevention of lipid metabolism alterations has been seen as a health priority.

In recent years, increasing evidence has shown the potential of natural products to modulate lipid cellular metabolism by the synergistic activity of their bioactive compounds on multiple molecular targets within a complementary medicine approach [4]. An attractive cosmopolitan medicinal mushroom is *Ganoderma lucidum* (*Gl*), which has been widely recognized by traditional medicine as a health promoter [5]. Previous studies have shown that triterpenes, polysaccharides, and proteoglycans isolated from *Gl* inhibit adipocyte differentiation [6], induce antidiabetic effects, and promote antihyperlipidemic and antioxidant activities [7]. Furthermore, *Gl* aqueous extracts show prebiotic properties and anti-inflammatory effects that can be used to control hyperlipidemia [8]. In fact, the role of *Gl* extracts as modulators of cholesterol metabolism has been recognized and already investigated in clinical trials [9]. In addition, research works have studied the modulation of genes and protein expression exerted by *Gl* [10,11], while others have focused on changes in microbiota composition [8,12]. Most reports have centered their attention on evaluating the expression of only a few pre-selected genes, while only a few have studied global expression changes in the liver to understand the effects of *Gl* for counteracting hypoglycemia [13] or neoplastic development [14]. However, none of them has addressed global transcriptomic changes and their association with physiological, cellular and gut microbiome conditions related to the hypocholesterolemic effects of *Gl*.

Accumulating evidence has also demonstrated that preparation, cultivation and manufacturing of *Ganoderma* products have a relevant impact on their medicinal properties. In the industry manufacturing fungal products, *Gl* has been cultivated on a variety of substrates and mixed with diverse supplements [1,2,3]. Furthermore, chemical constituents of mushrooms commonly differ among species or subspecies growing under differing environmental conditions. In Mexico, there is a great diversity of *Ganoderma* genetic resources, and several species have been identified at taxonomical and molecular levels [15].

Our group previously described the hypocholesterolemic properties of standardized and well-characterized hydroalcoholic extracts from the native *Ganoderma lucidum* strain CP-145 originating from the central region of Mexico. Molecular, taxonomic and biological data significantly substantiated that this strain has clear differences compared to other *Ganoderma* strains [15]. The consumption of *Gl* extracts by C57BL/6 mice fed with a high-cholesterol diet (HCD) significantly reduced total serum cholesterol, LDL, hepatic cholesterol, and triglycerides. *Gl* extracts also showed prebiotic effects, as they modulated the composition of gut microbiota by increasing the *Lactobacillaceae* family, particularly the genus *Lactobacillus* [16].

A novel approach is that one of our *Gl* extracts derives from basidiocarps cultivated in substrate containing acetylsalicylic acid (ASA). It has been shown that this drug exerts multiple effects on plant development and is capable of promoting the synthesis of secondary metabolites [17]. The effect of ASA on the functional and medicinal properties of *Gl* extracts has not yet been studied at a genomic level. In similar studies, the cultivation of *Agaricus bisporus* with ASA prevents post-harvest browning by maintaining the integrity of the membrane and facilitates the accumulation of phenols [18]. ASA supplementation on *Gl* enhances the levels of polysaccharides, triterpenes and ganoderic acid [19,20,21].

A more complete molecular characterization is needed to understand the underlying mechanisms of action of *Gl* on lipid metabolism, particularly for Mexican genetic resources under differing cultivation conditions. In this study, we performed whole gene expression analysis of liver and kidney tissues of C57BL/6 mice fed with a high-cholesterol diet (HCD) and the administration of standardized extracts from a Mexican *Gl* grown on a substrate with and without ASA. Then we explored correlations between RNA profiles established by the consumption of *Gl* extracts and biochemical features, as well as microbiota composition, aiming to provide a comprehensive view of molecular mechanisms linked to *Gl* hypocholesterolemic effects. On the basis of our results, we present a highly detailed landscape, previously unrecognized, of transcriptional changes occurring in the liver and kidney associated with the administration of *Gl* extracts. In both *Gl* extracts, we observed a positive enrichment in molecular programs relevant for the degradation and excretion of fatty acids. Importantly, we showed that the addition of ASA to the *Gl* cultivation substrate increases transcriptional responses that recapitulate those exerted by the simvastatin drug and are enriched to a higher degree in processes and pathways related to human obesity alterations in liver. We also revealed a profile of *Gl*-modulated genes that is associated with *Lactobacillus* enrichment and the reduction of cholesterol-derived fats in blood mediated by the *Gl* interventions. This study provides new evidence that adds to and extends the knowledge about the effects of natural resources, such as *Gl*, over transcriptional changes and the role of substrate composition for maximizing beneficial properties to prevent metabolic disorders associated with hypercholesterolemia.

## 2. Materials and Methods

### 2.1. Preparation of Standardized Mushroom Extracts

The strain CP-145 of *Gl* (GenBank accession number LN998989) was subcultured and incubated at 28–29 °C. Mushroom cultivation was carried out as previously described [10]. Polypropylene plastic bags (0.5 µm microfilter) were filled with oak (*Quercus acutifolia* Née) sawdust as substrate and prepared for mushroom cultivation: (a) *Gl*-1: 1300 mL of distilled water was added to a bag containing 1000 g of dry substrate; (b) *Gl*-2: acetylsalicylic acid (ASA, 10 mM; Sigma-Aldrich, St. Louis, Missouri, USA) in distilled water (1300 mL) was added to 1000 g of dry substrate. All bags were autoclaved, spawned, and incubated until complete mycelial colonization and fruiting. Hydroalcoholic extracts (32% by volume) of cultivated basidiocarps were processed in accordance to the patent MX322035-B [22]. Mushrooms extracts from dried basidiomata were concentrated and filter-sterilized, followed by freeze-drying overnight, and storing at 4 °C.

### 2.2. Biochemical Composition of Standardized *Ganoderma Lucidum (Gl)* Extracts

Biochemical composition of standardized *Gl* extracts, as well as the analytical methods and techniques performed to estimate each nutrient and organic profile, are described in Appendix A.

### 2.3. Animals and Treatments

Hypocholesterolemic and prebiotic effects of standardized *Gl* extracts were assessed in mice fed with a high-cholesterol diet, in comparison with the simvastatin drug. The effect of ASA, added to the substrate used for mushroom cultivation, on the functional and medicinal properties of *Gl* extracts was also studied (*Gl*-1: no ASA; *Gl*-2: 10 mM ASA). The concentrations of cholesterol (0.5%), simvastatin (0.03 g/100 g), as well as lyophilized *Gl*-1 and *Gl*-2 extracts (1.0%), in the animal diet were previously determined by the authors [16]. We used male 7-week-old C57BL/6 mice (26 g ± 0.5 g of weight) purchased from the animal facility of the Instituto Nacional de Ciencias Medicas y Nutrición Salvador Zubiran (INCMNSZ). Animals were housed (4/cage) in a 12-h light–dark cycle at a constant temperature (23 ± 2 °C) and relative humidity (45–55%). Mice were assigned to one of the following experimental groups (N = 40, n = 8 per experimental group) in a complete randomization strategy: (1) Control: Control diet (AIN-93); (2) HCD: High-cholesterol diet (0.5% *w*/*w* cholesterol; Sigma-Aldrich, St. Louis, Missouri, USA); (3) HCD + Sim: High-cholesterol diet (0.5%) plus simvastatin (0.03 g/100 g); (4) HCD + *Gl*-1: High-cholesterol diet (0.5%) plus the *Gl*-1 extract (1.0% *w*/*w*); and (5) HCD + *Gl*-2: High-cholesterol diet (0.5%) plus the *Gl*-2 extract (1.0% *w*/*w*). Concentrations studied of cholesterol, simvastatin or *Gl* extracts were added to the food pellet and mixed homogeneously, depending on each experimental treatment. Animals were fed ad libitum with AIN-93 standard and experimental diets (average intake: 3.47 ± 0.22 g/day). Water was also provided ad libitum. Sample size was calculated in accordance with the 3R’s model with G*power software (two tail test, power = 0.95 and effect size = 2). Allocation and treatment were performed with a not blinded strategy. At day 43, mice were deprived of food and water for 8 h, and blood samples from the portal vein were collected for serological analysis. Mice were sacrificed at this end point, and liver and kidney tissues were collected and stored at −80 °C for further processing. Animal studies were approved by Ethics Review Committee at the INCMNSZ (FNU-1180-15/16-1). Diet composition and treatments are described in Appendix A.

### 2.4. RNA Extraction

RNA was isolated with TRIzol reagent (Invitrogen, Waltham, MA, USA), according to the manufacturer’s protocol. RNA concentration was assessed with Nanodrop 2000 spectrophotometer and RNA quality was evaluated by the Bioanalyzer system (Agilent, Santa Clara, CA, USA). Samples from each experimental condition were pooled in equimolar concentrations (80 μg/μL) in A (N = 4), B (N = 4), C model, where C is A + B.

### 2.5. Gene Expression Profiles

We hybridized 4000 μg of each RNA pool to the mouse gene st v1.0 (Affymetrix, Waltham, MA, USA) microarrays and were successively washed and scanned according to manufacturer’s instructions. Briefly, pool total RNA was converted to first-strand cDNA using Superscript II reverse transcriptase primed by a poly(T) oligomer. This was followed by a cDNA synthesis to generate cRNA products, that were subsequently fragmented into 200 nucleotides. Array hybridization and washing were then performed. Finally, the GeneChip Scanner 3000 7G (Affymetrix, Waltham, MA, USA) was used to collect fluorescence signal.

### 2.6. Bioinformatic Analysis

mRNA signal intensities were background corrected by RMA and quantile normalized using the Bioconductor library oligo [23] in the R environment. Batch effect was attenuated with the ComBat algorithm [24], on Genepattern [25,26]. Probesets were annotated with mogene10sttranscriptcluster.db library of Bioconductor, and unassigned probes were removed. Duplicated probes were filtered by selecting the probe with the highest interquartile range. Differential expression was computed by an adjusted moderate *t*-test implemented in the limma package [27] on R. Genes with an absolute log fold change > 0.7 and adjusted *p* value ≤ 0.05 were considered as differentially expressed (DE). Functional annotations for DE genes were retrieved from WebGestalt [28,29], with the Biological process, Kegg and Reactome annotations. Gene annotations with a *p* value lower than 0.05 were retained significant. Gene expression profiles are available through GEO omnibus with the accession GSE159656. Raw human liver data were acquired from GEO (GSE15653) and were processed as previously described in the R environment. Human homologues of DE genes were retrieved from HomoloGene [30].

### 2.7. Gene-Targets and Hypercholesterolemia Associations

The Open Targets Platform website [31,32], to prioritize and identify the gene targets associated with hypercholesterolemia. The Open Targets Platform score target-disease associations based on evidence from six levels: genetics, genomics, transcriptomics, drugs, animal models, and scientific literature. The complete significant differentially expressed genes of each contrast (*Gl*-1, *Gl*-2, simvastatin vs. HCD) were included in the analysis. The reported overall score is the sum of the individual data source scores.

### 2.8. Evaluation of Toxicity Profiles

Public gene expression data from DrugMatrix [33] and Open TG-GATEs [34] were downloaded from the ToxicoDB data base [35] including gene expression profiles from: Open TG-GATEs Rat, Open TG-GATEs Human and DrugMatrix Rat. Both databases fuse compound-induced gene expression data from hepatic tissue of in vivo models treated with safe and toxic doses. We only include data from a high dose and 24 h of treatment condition, only those genes with a logFc ≥ 1 and false discovery rate (FDR) ≤ 0.05 were selected for transcriptional signatures. Additionally, well-known Reactome pathways related with drug toxicity were downloaded from the Molecular Signatures Database v7.2 (MSigDB, [36,37]), and single sample gene set enrichment analysis (ssGSEA) was computed on GSVA R package based on the above-described data and normalized gene expression from our experimental conditions to computed toxicity-scores.

### 2.9. Identification of Transcription Factor (TF) and Selection of Putative Target Binding Sites in Genes of Interest

DE genes annotated as DNA-binding transcription factors were identified using the TFcheckpoint data base [38]. Thereafter, Pearson correlation between predicted TF and DE genes was computed along treatments. Significant correlations were considered with a Pearson coefficient > 0.7 and adjusted *p* value < 0.2. DNA binding motifs in the promoter sequence of genes, significantly correlated with each TF, were analyzed with MotifDb and Biomart packages of Bioconductor and the JASPAR matrices database [39,40], in order to support the evidence on the discovery of bona fide TF gene targets. Binding sites with a minimum score identity of 80% were retained. Network visualization was done with Cytoscape software [41].

### 2.10. Defining Drug–Target Interactions

Known simvastatin-modulated genes were retrieved from the DGIdb database [42,43]. To identify potentially druggable genes among *Gl*-modulated genes, the DGIdb database and the Drug Repurposing Hub [44,45] were consulted.

### 2.11. Cell Line Models

RAW 264.7 (ATCC) macrophage-derived murine cell lines were maintained in Dulbecco’s Modified Eagle Medium (DMEM, Gibco, Waltham, MA, USA) medium supplemented with 10% fetal bovine serum (Gibco, Waltham, MA, USA) at 37 °C in 5% CO_2_.

### 2.12. Lipid Stained with Oil Red

RAW 264.7 cells were treated with *Gl*-1 and *Gl*-2 extracts for 48 h and cultured under high-cholesterol concentration (50 µM) for 72 h (Gibco, Waltham, MA, USA). Cells were stained with Oil Red O (0.3% *w*/*v*; Sigma-Aldrich, St. Louis, MO, USA) for 30 min at room temperature. We took images for relative quantification. Color was dissolved in 100% isopropanol and the absorbance was measured spectrophotometrically with a BioRad apparatus. Extra wells were stained with sulforhodamine B (SRB) to normalize against the total number of cells per condition.

### 2.13. Real-Time Polymerase Chain Reaction (RT-PCR) and Western Blot (WB) Analysis

Real-time polymerase chain reaction (RT-PCR) was carried out using TaqMan Gene Expression Assays (Applied Biosystems, Waltham, MA, USA), and GAPDH as an endogenous control. Delta-Ct values were computed with the HTqPCR library on R. All the experiments were performed in triplicate on a Step One Plus system (Applied Biosystems, Waltham, MA, USA) and data were processed by a DeltaCt method. Proteins were extracted with RIPA lysis buffer and 10 mg/μL were assessed in triplicate by Western Blot and detected against FASN (ab22759, 1:2000, Abcam, UK) and ELOVL6 (ab69857, 1:1000, Abcam, UK). Proteins were visualized by enhanced chemiluminescence detection system (GE Healthcare, Chicago, IL, USA). Densitometry analysis was performed with ImageJ software.

### 2.14. M1/M2 In Vitro Polarization

On RAW 264.7 cultivation Interferon gamma (INFy) (1 mg/mL) was added to stimulate M1 polarization and IL-4 (20 ng/mL) for M2 polarization during 24 h. Polarization markers (Nos2, Irf5, Stat1, Stat6, Irf4) were then evaluated by RT-PCR as described above.

### 2.15. Gut Microbiota Analysis

DNA from mice fecal samples collected during the last week of the in vivo experiment was isolated with QIAamp DNA Stool Mini Kit (Qiagen, Venlo, The Netherlands), according to the manufacturer’s protocol. Variable V3 and V4 regions of the 16S rRNA were amplified from 2.5 μL of microbial genomic DNA (5 ng/μL) with dedicated primers (Forward:5′-AAACTCAAAKGAATTGACGG-3′, 61.2 °C; reverse: 5’-CTCACRRCACGAGCTGAC-3′, 56.9 °C) and KAPA HiFi DNA Polymerase (Roche, Basel, Switzerland), containing the Illumina adapter sequences for library preparation for an amplicon of ~550 bp. The V3 and V4 regions were purified by Ampure XP (Beckman Coulter, Brea, CA, USA) and quantified on an Agilent 2100 Bioanalyzer system (Agilent Technologies, Santa Clara, CA, USA). Dual-index barcodes were attached by a PCR reaction (12.5 ng in 2.5 μL of the V3 and V4 regions and two 5 μL of each Index primer) with KAPA HiFi HotStart ReadyMix (Roche, Basel, Switzerland) in accordance with procedures for 16S Ribosomal RNA Gene Amplicons for the Illumina MiSeq System guide of Illumina. Final amplicons (~610 bp) were cleaned with Ampure XP bits, quantified on the Agilent 2100 Bioanalyzer and sequencing on Illumina MiSeq platform (MiSeq Reagent Kit V.3, 600 cycles), according to the manufacturer’s instructions using paired 300-bp reads. Overlapping paired-end reads sequencings were merged by fastq-join. Microbiome analysis from raw DNA fastq were performed with QIIME V.1.9 [46]. Only Illumina reads with an average score above 20 were retained for further analysis. Assignment of an operational taxonomic unit (OTU) of each amplicon was generated at a variety of similarity thresholds (99.9% for phylum, 99.7% for class, 99.6% for order, 90.4% for family, 69.7% for genus) using the USEARCH V5.2.236 [47]. The Shannon diversity index was computed on R software environment based on species frequency data from our previous study [21], as H = −sum pi * ln(pi), where pi is the proportional abundance of species i.

### 2.16. Correlation Analysis

To estimate the association between transcriptome changes and abundances of particular bacterial taxa, paired Pearson’s correlation coefficient and adjusted *p* values (Benjamini–Hochberg) were computed in the R software environment. Pearson coefficient > ±0.5 and adjusted *p* value < 0.1 were considered as significant.

### 2.17. Gene Over-Representation Analysis

Each set of genes significantly associated with particular bacterial taxa was used for performing a hypergeometric test to identify significant enrichment of Gene Ontology (GO) biological processes terms, cellular component and molecular function categories. We used the ConsensusPath Db online tool (http://cpdb.molgen.mpg.de/) for that purpose. Enrichments having adjusted *p* value < 0.1 were retained for analysis.

### 2.18. Blood Systemic Parameters

COBAS C111 analyzer (Roche Diagnostics Ltd., Basel, Switzerland) was used to assess serum concentrations of total cholesterol (TC), total triglycerides (TG), high-density lipoprotein cholesterol (HDL), and low-density lipoprotein cholesterol (LDL), as previously described [48].

### 2.19. Statistical Analysis

Statistical differences across the experimental conditions were computed by a non-parametric analysis using Wilcoxon or Kruskal–Wallis tests, to compare two groups or multiple groups, respectively, using R software. A *p*-value ≤ 5% was considered significant. *p*-value: * <= 0.05, ** <= 0.01, *** <= 0.001, **** <= 0.0001. For microarray analysis, a moderated *t*-test was performed and FDR was applied to adjusted *p* values for multiple hypothesis on limma library of the Bioconductor/R environment.

## 3. Results

### 3.1. A High-Cholesterol Diet (HCD)-Altered Lipid Metabolism in Liver and Kidney of C57BL/6 Mice

Under normal physiological conditions, fatty acid (FA) metabolism involves a fine-tuned balance between anabolic and catabolic signaling that results in the production of important molecules and substrates for cellular function and homeostasis (Figure 1A) [49,50,51,52,53,54]. Differential expression profiles associated with HCD were identified in liver and kidney tissues in vivo (Appendix A). Enrichment pathway analysis (EPA) of altered hepatic genes under HCD, in comparison to standard diet (SD), revealed effects on relevant pathways including biosynthesis of FA and cholesterol, as well as mobilization and transport of lipids (Appendix A). In addition, a set of keratins were up-modulated in response to HCD, which may affect mitochondrial functions and inflammatory responses. Kidney gene profiling was associated with the up-modulation of steroid and lipid metabolic processes in the HCD group (Appendix A). These results are in agreement with the fact that high cholesterol conditions induce a reprogramming of lipid metabolism resulting in hypercholesterolemia. The hepatic aberrant activation of small metabolic molecules and up-modulation of ABC transporters leads to cellular incorporation of lipids (Appendix A).

### 3.2. Ganoderma Lucidum Extracts Regulated Lipogenic and Metabolic Signaling Pathways in the Liver

Global transcriptional changes in liver tissues of mice fed with a HCD and treated with standardized *Gl* extracts were analyzed (Appendix A). EPA of differentially expressed (DE) genes (logFC > 0.7 or < −0.7, adjusted *p* value < 0.05) responsive to *Gl* treatment revealed that lipid biosynthesis, de novo lipogenesis and triglyceride metabolism were impaired (Figure 1B). We previously reported that *Gl* extracts reduced the expression of relevant genes involved in cholesterol biosynthesis, such as Acss2, Acly, Fdps, Nsdhl, Acaca, and Fasn, which were probably silenced by a decrease of the Srebf1/Srebp1 transcriptional network (Figure 1C,D, Figure 2A and Figure 3A).

Our profile data also revealed that the administration of standardized *Gl* extracts induces the expression of genes involved in FA oxidation (FAO), lipolysis, and bile acid synthesis pathways. Furthermore, *Gl* extracts increased FAO via the induction of peroxisome proliferator-activated receptors (PPARs), normally suppressed in obesity and fatty liver. These results suggest that the *Gl*-1 extract triggers the PPAR route activation by promoting the interaction between PPAR and E2fs transcriptional factors, specifically through the up-modulation of E2f8 (Figure 2A). By contrast, the *Gl*-2 extract improved PPAR activity by increasing the cytochrome P450 family enzymes, specifically Cyp4a, resulting in increased FAO (Figure 3A). PPAR activation was consequently associated to the up-modulation of target genes, such as the early growth response 1/2 (Egr1/2) and the AP-1 transcription factor subunit (Figure 3B). Moreover, several mitogen-activated protein kinase (MAPK) family genes (e.g., Fos, Jun, Il1a, Il1b) were over-expressed after *Gl*-2 extract treatment (Figure 3B).

Gene enrichment analysis also showed that another effector of standardized *Gl* extracts in animals consuming a HCD is the increased synthesis of bile acids (BAs). The *Gl*-1 extract triggered one of the major bile acids synthesis pathways via the Cyp7b1-mediated hydroxylation of cholesterol (Figure 2A). By contrast, the *Gl*-2 extract increased leukotrienes through Cyp4a expression, stimulating the mechanism of bile acids production, biotransformation and excretion (Figure 3A).

Standardized *Gl* extracts also prevented endoplasmic reticulum stress and protein misfolding by improving pathways, such as Derl3 and the heat shock protein genes Hsp1a/b and Dnajb1 (Figure 2A and Figure 3A). The gene Saa2 was up-modulated by the *Gl*-1 extract, that might improve intracellular trafficking of cholesterol, as a means to avoid its accumulation (Figure 2B). Another relevant gene required to mobilize cholesterol outside the cell is the transcriptional regulator Atf3, which was up-modulated by the *Gl*-2 extract (Figure 3A).

Our transcriptional data showed relevant mechanisms by which standardized *Gl* extracts prevent hepatic lipid accumulation through significantly decreasing lipogenesis programs, and restoring lipid homeostasis (lipid processing, reverse cholesterol transport and lipid export), which consequently enhances catabolic pathways over anabolic pathways (Figure 1, Figure 2 and Figure 3; Appendix A).

### 3.3. Standardized Gl Extracts Recapitulated Simvastatin Analogous Mechanisms in Liver Tissue

We then analyzed the expression profiles associated to simvastatin treatment under a HCD. As expected, EPA revealed the significant down-modulation (logFC > 0.7 or < −0.7, adjusted *p* value < 0.05) of biological processes involved in energy homeostasis (Appendix A), such as fatty Acyl-CoA biosynthesis, triglyceride biosynthesis, lipid metabolic process, among others. Moreover, simvastatin treatment robustly enriched the expression of lipid catabolic processes, fat digestion and absorption, PPAR signaling pathways, and fatty acid degradation mechanism, achieved by the activation of Pnlip, Clps, Pnliprp1, and Acot3, key lipolytic enzymes (Figure 1B and Appendix A; Appendix A).

Additional effects of simvastatin actionable pathways include anti-inflammatory properties, and up-modulation of cellular stress responses. Atp2a1 and Derl3 genes were also up-regulated in response to high cholesterol stress enhancing the efficiency of the endoplasmic reticulum system (Appendix A**)**. Rab30, which increased exocytosis of cholesterol vesicles [55], was also over-expressed (Appendix A). Simvastatin enhanced membrane-cytoskeleton adhesion through the up-modulation of the cytoskeleton related-genes Myh1, Myh2, Myl1 and Myl2, thus modulating endocytosis, exocytosis and cell adhesion (Appendix A).

Standardized *Gl* extracts (*Gl*-1, *Gl*-2) and simvastatin treatments exhibited heterogeneous gene expression patterns. However, they displayed similar enrichment of well-known lipid metabolic pathways, thus their hypocholesterolemic properties modulate equivalent signaling mechanisms (Appendix A).

Using the Open Targets Platform, which systematically integrates multiple biomedical data to associate genes with diseases, we determined the relative association of *Gl* and simvastatin-modulated genes with hypercholesterolemia condition. Interestingly, *Gl*-2 gene-targets are more robustly linked to the disease than *Gl*-1 targets (Appendix A). Further similar relevance scores, which described the probability that a set of genes are specific to a particular disease, were observed between *Gl*-2 and simvastatin (Appendix A). In line with these results, by exploring the DGIdb database we identified several well-described pharmacological target genes of simvastatin, which were also modulated by *Gl* (Figure 4A). These data suggest that down-expression of de novo lipid synthesis genes and the up-regulation of relevant enzymes related to lipid elimination are critical for simvastatin and *Gl* extracts mediated by analogous mechanisms. In general, *Gl* extracts, mainly *Gl*-2 and simvastatin promote common transcriptional networks and phenotypic responses to manage high hepatic cholesterol concentrations, which have potential pharmacological properties. We finally interrogated gene-expression profiles of toxicity by computing ssGSEA scores based on transcriptional signatures from DrugMatrix and Open TG-GATEs, as well as toxicity-related pathways. Non-significant changes in the hepatic toxicity portraits were observed between *Gl*-1, *Gl*-2 and simvastatin. These data suggest that the use of *Gl* extracts at the administrated doses is safe and no toxicity effects on hepatic cells were greater than those defined in the tissue from mice treated with the allopathic simvastatin drug (Appendix A, Appendix A).

### 3.4. Unique Hepatic Transcriptional-Pathway Portraits Derived from Gl-1 and Gl-2 Extracts

There were a substantial number of transcripts whose expression was specific to each treatment, as shown in the Venn diagram (Appendix A). For instance, the *Gl*-2 extract established a higher number of DE genes. We performed an EPA in order to examine whether there was a recognizable biological relevance of differential expression patterns between *Gl*-1 versus *Gl*-2 extracts. Identified altered genes were grouped in diverse functional categories, including response to hormone and cAMP, estrogen signaling pathway, and the activation of Ap-1 family of transcription factors (Appendix A). These differences are explained by differing bioactive compounds contained in each extract. We showed that the ASA (10 mM) added to the substrate used for mushroom cultivation, not only changed the chemical composition of the *Gl*-2 extract, in comparison to the control (*Gl*-1, no ASA), but also its functional and medicinal properties. Gene expression profiles of *Gl* extracts versus simvastatin treatments led to heterogeneous transcriptional portraits, which are enriched in post-transcriptional protein folding and digestion pathways (Appendix A).

Relevantly, the most common gene classes modulated by *Gl* extracts were enzymes, transporters, transcription factors (TFs), oxidoreductases, and transferases (Figure 4B,C). These findings showed that, although treatments with *Gl* extracts and simvastatin did not generate the same transcriptional portraits in mice fed with a HCD, they definitely induced common cellular signaling processes capable of protecting against high lipid concentrations.

### 3.5. Gl-2 Extract Led to the Establishment of Dedicated Hepatic Transcriptional Factor Interaction Networks

The elucidation of *Gl*-mediated transcriptional factor (TFs) and their target genes is crucial to understand the anti-hypercholesterolemic mechanisms of *Gl*-2 extract. We identified many potential TFs, including Npas2 (component of the Srebp complex), which were down-modulated, whereas Egr1, Egr2, Junb, Nr4a1, Fos and Jun were up-modulated under *Gl*-2 intervention (Figure 4D). A set of DE genes positively correlated with predicted TFs was identified (Pearson correlation > 0.7, adjusted *p* value < 0.2). Enrichment of TF domains in promoter regions of correlated genes was evaluated in order to define correlated genes and predicted TFs, as well as to identify possible downstream targets (Figure 4D). The network of *Gl*-2 modulated TFs and target genes showed the down-modulation of carbon metabolism, FA elongation and metabolism, *Gl*ycolysis/*Gl*uconeogenesis and lipid biosynthetic process, as well as a gain of function of MAPK signaling pathway, and estrogen pathway activation of the AP-1 family (Figure 4D,E). Collectively, these observations suggest that transcriptional regulation induced by the *Gl*-2 extract is in part explained by the altered expression of TFs.

### 3.6. Effect of Gl Extracts on Macrophages and Cholesterol Homeostasis

The liver represents the major reservoir of macrophages among all solid organs [56]. Interestingly, liver macrophages have been shown to be instrumental in hepatic metabolic responses, apart from their immunological activity. We evaluated the RAW 264.7 macrophage cell line, which shows similarities to murine hepatic macrophages [57], to investigate underlying mechanisms of *Gl* effects on cholesterol metabolism in hepatic macrophages. RAW 264.7 macrophages were treated with *Gl* extracts cultured under high cholesterol concentrations (50 µM). Treated RAW 264.7 macrophages were then subjected to oil red O staining for investigating lipid accumulation. Data showed that cells cultured on a medium containing a high cholesterol concentration and fungal extracts (*Gl*-1, *Gl*-2), were less enriched in lipid accumulation than those RAW 264.7 cells cultured in the control condition (Figure 5A,B). Our results also showed that treatments with *Gl* extracts significantly attenuate the expression of Fasn and Elovl6 at protein level (*p* < 0.05 vs. Control, Figure 5C), which are associated with fatty acid synthesis, and they were also abolished by *Gl* extracts in hepatic tissue. Macrophage polarization markers were also evaluated in order to assess whether hepatic immunity state is affected by *Gl* extracts and cholesterol consumption. Cholesterol loaded-macrophages show different expression profiles compared to non-polarized macrophages (M0) (Appendix A). We found an ambiguous phenotype between M1/M2-activated macrophages among *Gl* treatments and cholesterol. A decrease in the pro-inflammatory M1 markers Nos2 and Stat1 was detected in mice treated with *Gl* extracts and high cholesterol consumption compared to cholesterol loaded-macrophages, which is consistent with the anti-inflammatory properties of *Gl* (Appendix A).

### 3.7. Hepatic Transcriptional Reprogramming by Gl Extracts in Mice Fed with a HCD Were Consistent with an Altered Liver Landscape from Obese Patients

We assessed whether our results on the altered transcriptional landscape on murine liver tissues from animals consuming *Gl* extracts and a HCD had biological significance in a human context (Figure 5D,E). *Gl* target genes were compared with public human gene expression profiles of surgical liver biopsies from 13 obese and 5 control subjects (GSE15653). Most cellular signaling modulated by *Gl* extracts (*Gl*-1, *Gl*-2) in the mouse model were also altered by hepatic lipid accumulation in obese human patients. Interestingly, genes differentially expressed by *Gl*-2 show a higher enrichment in pathways, such as FA, carbon and retinol metabolisms; PPAR signaling and bile acid pathways; among others (Figure 5E). These data suggest potential regulation effects of *Gl* extracts to exert equivalent hypocholesterolemic effects on human liver tissue having high cholesterol levels.

### 3.8. Renal Transcriptional Landscape of Mice Fed with *Gl* Extracts and a HCD

Gene and pathway expression portraits were assessed to understand underlying mechanisms of *Gl* extracts and simvastatin effects under metabolic stress of kidney. An increased number of altered genes was detected in the *Gl*-2 extract treatment (Appendix A). Renal transcriptional analysis of *Gl* extracts consumption under a HCD showed a moderate enrichment of signaling processes related to lipid pathways. This was so for FA responses, isoprenoid biosynthetic process, and steroid metabolism for the *Gl*-1 extract intervention, whereas PPAR signaling pathways and cellular responses to lipid and hormones were enriched after the *Gl*-2 extract intervention (Figure 5F; Appendix A). In addition to lipid metabolism, the *Gl*-2 extract also had an impact on the inflammatory mediator regulation of TRP channels, MAPK signaling pathways, activation to AP-1 TFs, regulation of cell death, among others (Figure 5F). These data revealed novel signaling circuits associated to *Gl* extracts properties on kidney.

Enriched pathways were consistent in both liver and kidney tissues showing an enhanced anabolic cholesterol metabolism, although the concordance between hepatic and renal tissue was limited at gene level (Appendix A). Unexpectedly, we did not detect any significantly enriched pathway of mice treated with simvastatin. This can be a consequence of the limited number of DE genes listed as significant, as well as a more specific activity of statins on hepatic tissue (Appendix A).

### 3.9. *Gl* Extracts Consumption Promoted Bacterial Richness in the Gut Microbiota

Diversity of bacteria in the gut microbiota, estimated as the number of species present and the Shannon index, was similar in mice fed with a HCD + *Gl* extracts, HCD + simvastatin and the Control group, but it was greater than mice fed with a HCD only (Figure 6A). These results indicated that *Gl* dietary intervention increased species richness, showing bacterial diversity similar to the control group (Figure 6B). This confirms our results previously reported showing that more than 90% of the relative abundance in all experimental groups was represented by three phyla: Bacteroidetes, Firmicutes, and Proteobacteria [16]. The mice group consuming a HCD showed dysbiosis, as *Bacteroides acidifaciens*, *Mucispirillum schaedleri*, and *Parabacteroides distasonis* increased significantly, while *Faecalibacterium prausnitzii* and *Prevotella copri* had a significant decrease. This dysbiosis was positively reversed by *Gl* extracts (Figure 6B). Furthermore, the consumption of *Gl* extracts greatly increased the abundance of *Lactobacillus*, which is part of the Firmicutes phylum (Figure 6C,D), so we focused further analyses on this genus.

### 3.10. Relevant Target Genes of Gl Extracts Were Correlated with Blood Lipids and *Lactobacillus* Abundance in the Gut Microbiota

Correlation analyses were performed between gene expression changes and the levels of total triglycerides (TG), low-density lipoprotein cholesterol (LDL), and high-density lipoprotein cholesterol (HDL) in experimental treatments, in order to better characterize transcriptional changes induced by *Gl* extracts (*Gl*-1, *Gl*-2) within the clinical context of lipid metabolism alterations.

Both *Gl*-1 and *Gl*-2 extracts produced a significant reduction of all blood lipids evaluated (Figure 6C,D), while HDL/LDL ratio were almost restore to normal levels by *Gl* extracts (Appendix A). We found a set of genes positively or negatively correlated (R ≥ 0.5, adjusted *p* ≤ 0.1) with TG, LDL, and HDL levels in each *Gl* treatment (Figure 6C,D; Appendix A). In general, a greater number of responsive genes to the *Gl*-2 extract were significantly correlated with any of the evaluated biochemical variables, in comparison with responsive genes to the *Gl*-1 extract. A similar approach was followed to explore the association between genes responsive to *Gl*-1 or *Gl*-2 extracts and the gut microbiota composition. We identified a group of *Gl*-responsive genes which are correlated with the increased abundance of the genus *Lactobacillus* in the gut microbiota (R ≥ 0.5, adjusted *p* ≤ 0.1) (Figure 6C,D; Appendix A). Remarkably, 80.7% (*Gl*-1) and 98.4% (*Gl*-2) of responsive genes, showing significant correlation with *Lactobacillus* abundance, were also correlated with at least one biochemical variable under the same *Gl* treatment. The biological function of *Gl*-correlated genes from each extract was elucidated through pathway enrichment analysis. As expected, top pathways were FA metabolism and the inflammatory immune response (Appendix A). These results strongly suggest that *Gl* extracts exert an effect on the expression of genes acting on lipid metabolism pathways, partly through modulation of the gut microbiota.

## 4. Discussion

Accumulating evidence from previous biological and pharmacological research demonstrated that *Gl* extracts and their bioactive compounds show anti-hypercholesterolemic, hypoglycemic [6], and antioxidant activities [7], and modulate the gut microbiota [8]. However, most studies only evaluate a limited set of genes, and it remains unknown how *G. lucidum* modulates whole cell transcriptional-programs. To the best of our knowledge, this study presents the first comprehensive transcriptional landscape of *Gl*-modulated genes in hepatic and renal tissue of a murine model under HCD, which allows us to propose how standardized extracts from Mexican *Gl* operate on and identify unique molecular mechanisms, treatment-targets, similarities to the lowering-cholesterol simvastatin drug mechanisms, and correlations with relevant circulating biochemical parameters and gut microbiota composition, including differences across *Gl* extracts obtained from basidiocarps cultivated on different substrates.

Metabolic fluxes are altered under excess of cholesterol and fatty acids, leading to metabolic disorders and chronic inflammation [58]. We described the signal transduction pathways established by the consumption of *Gl* extracts (*Gl*-1, *Gl*-2), under a HCD, accounting for a metabolic reprogramming of hepatic, renal and macrophage cells from in vivo and in vitro models. Our study showed that Mexican *Gl* extracts prevent FA synthesis and accumulation through down-modulation of genes involved in lipogenesis, elongation and desaturation. Furthermore, after the consumption of *Gl* extracts, there was an enhancement of FA degradation and excretion, through the activation of peroxisome proliferator-activated receptors (PPARs), fatty acid oxidation and bile acid conversion.

The activation of FAO and liver X receptors mechanisms (LXRs), a nuclear receptor family of transcription factors that contribute to cholesterol homeostasis by stimulating cholesterol conversion to bile acids [3,4], reduce plasma cholesterol levels and raise plasma HDL levels in mice [59,60,61]. In agreement with all these data, the measurement of biochemical blood parameters, such as LDL and TG, showed a decrease after the consumption of *Gl* extracts by our animal model. Exploratory clinical trials confirmed these results [9]. Several defense mechanisms are activated for maintaining protein homeostasis under stress conditions, such as hypercholesterolemia. The endoplasmic reticulum (ER) plays an important role in regulating lipid metabolism, protein synthesis, protein post-translational modifications and trafficking [62]. Moreover, lipid regulatory molecules, such as cholesterol, can be involved in the generation of misfolded proteins and ER stress, particularly in lipid raft areas, which are membrane domains rich in cholesterol and sphingolipids. Standardized *Gl* extracts prevented ER stress and protein misfolding by improving different pathways preventing cell damage due to high cholesterol concentrations.

We also showed that metabolic gene expression profiles are correlated to biochemical parameters, indicating that these changes occur in a context-dependent manner. The treatment with *Gl* extracts establishes a pathway landscape analogous to that from simvastatin. Thus *Gl* extracts counteract dietary-induced changes caused by a HCD intervention, prompting catabolic over anabolic pathways. As lipid metabolism is a complex multi-organ process, we considered that renal transcriptional metabolic portraits could also account for the *Gl* extracts effects on FA and cholesterol mechanisms, and their surrogate signaling pathways. Overall, kidney gene profiles from mice fed with a HCD and treated with *Gl* extracts were consistent with hepatic gene expression signatures, integrating an enhanced catabolic cholesterol metabolism.

There have been no previous studies on the transcriptional effect of *Gl* extracts obtained from basidiocarps cultivated on substrate containing ASA. Our data demonstrated that specific mechanisms exist for *Gl*-2 extracts, such as improved modulation of gene-expression programs, probably established by a fine tuning of core transactional factor networks that lead to a more complex transcriptional effect. Moreover, we detected more robust similarities between *Gl*-2 and simvastatin gene-expression activity. Our recent study demonstrated that the addition of ASA in the *Gl* cultivation substrate stimulates the production of more or new bioactive compounds, mainly the polysaccharides, B-methyl glucopyranoside, DL-arabinitol and ribitol, in comparison with the conventional extract (no ASA) [63]. This pattern might be the result of biotransformation processes that occur when fungal enzymes catalyze chemical modifications of a compound [64]. From a biomedical point of view, we focused on the acetylsalicylic acid, a non-steroidal anti-inflammatory drug approved for the treatment of pain and fever, that has also been included as a secondary prevention in stroke treatment and cardiovascular disease at low dose. ASA has also demonstrated improved effects when administrated in combination with other cholesterol-lowering drugs, such as simvastatin. Coadministration of both drugs reduces the incidence of cardiovascular disease and improves blood circulation [65,66]. A proposed means to obtain this synergistic effect is probably by an improved solubility of simvastatin by ASA that consequently enhances its bioavailability [67]. How ASA was biotransformed by *Gl* or how ASA non-converted compounds might maximize the *Gl*-2 extract needs to be further addressed.

In line with our predicted toxicity analysis based on gene expression portraits, our previous study on an in-vivo toxicity assay on rat models did not show significant toxicity events at different doses, including the dose used in this study. Thus, the standardized hydroalcoholic extracts of Mexican *G. lucidum* are safe and no significant adverse, toxic or harmful effects have been recorded (data not shown, manuscript in preparation), making our *Gl* extracts an attractive coadjuvant treatment option.

Furthermore, recent evidence suggested a strong correlation between immune and metabolic systems [68], as well as coordinated biological regulation mechanisms [69]. Macrophages are recognized as relevant cells in metabolic pathologies through the induction of inflammation and foam cell formation (lipid accumulation) in response to high cholesterol concentrations [70]. Our findings showed that the treatment with *Gl* extracts significantly contributes to change macrophage activity. In vitro treatment of hepatic-like macrophages with *Gl* extracts (*Gl*-1, *Gl*-2) reduced de novo lipogenesis and lipid accumulation, in accordance with previous reports [8]. Furthermore, macrophages react to various environmental stimuli, which can modify their phenotypes [71]. The increase of non-HDL cholesterol has been linked to a greater monocyte recruitment, and a subsequent polarization to M1 macrophages [72]. In this study, we demonstrated a partial reversion of the phenotype established by high cholesterol concentrations via the administration *Gl* extracts, through the down-modulation of the M1-associated markers Nos2 and Stat1.

In addition to gene modulation, our in vivo experiments showed that the administration of *Gl* extracts also modulates the composition of the gut microbiota, triggering an increase of the Lactobacillaceae family and the genus *Lactobacillus*, as a result of their bioactive compounds, mainly α-glucans and β-glucans. Additionally, low species diversity has been linked to the establishment and development of diverse human pathologies, such as obesity [73]. We observed an increase of species diversity after the intervention with *Gl* extracts under a HCD, compared to the mice group fed with a HCD only, restoring microbiome richness and diversity under a HCD. Overall results indicate that beneficial effects of consuming *Gl* extracts may be partly attributed to the increase of *Lactobacillus* in the gut, through regulation of transcriptional programs triggering a catabolic phenotype.

Potential sources of variability in product chemistry and efficacies need to be addressed in natural products manufacturing [74]. In this context, we confront this challenge by studying well characterized patented *Gl* extracts, and confirming the molecular identification of Mexican *Ganoderma lucidum*. This has been a relevant bias in other studies that analyzed commercial extracts labeled as *Ganoderma* species that are not correctly assigned [74]. We characterized the strain studied of *Ganoderma lucidum* by sequencing the ITS1-5.8S-ITS2 region from the rDNA, showing that it clearly separated in the consensus the phylogenetic trees of reference strains from European, North American and Southeast Asian origin [15]. Our work therefore shows new data from an independent genotype on how cellular expression portraits of hepatic and renal tissues are modulated in response to *Gl* extracts treatment and a HCD diet. We recognized well-known driver *Gl*-target genes, as well as novel contributing transcriptional circuits not previously described well due to limited high-throughput information.

## 5. Conclusions

Our preclinical evidence notably shows the first deep high-throughput genomic characterization of the mechanisms of action of *Gl* extracts under a HCD. Collectively, our data show that the consumption of *Gl* extracts brings about diverse potential health benefits: (1) to improve gene expression programs dedicated to FA degradation and excretion; and (2) to modulate the gut microbiota for enhancing prebiotic and microbiota-modulating transcriptional effects. These processes contribute synergistically to reduce cholesterol accumulation and inflammation. Understanding the molecular pathways that take place under *Gl* extracts consumption will provide new insights into the hypocholesterolemic and prebiotic properties exerted by *Gl*. The exciting progress in this field supports the promising use of a robust complementary medicine to benefit patients with hyperlipidemia through standardized and well-characterized *Gl* extracts.

## Figures and Tables

**Figure 1 nutrients-13-00038-f001:**
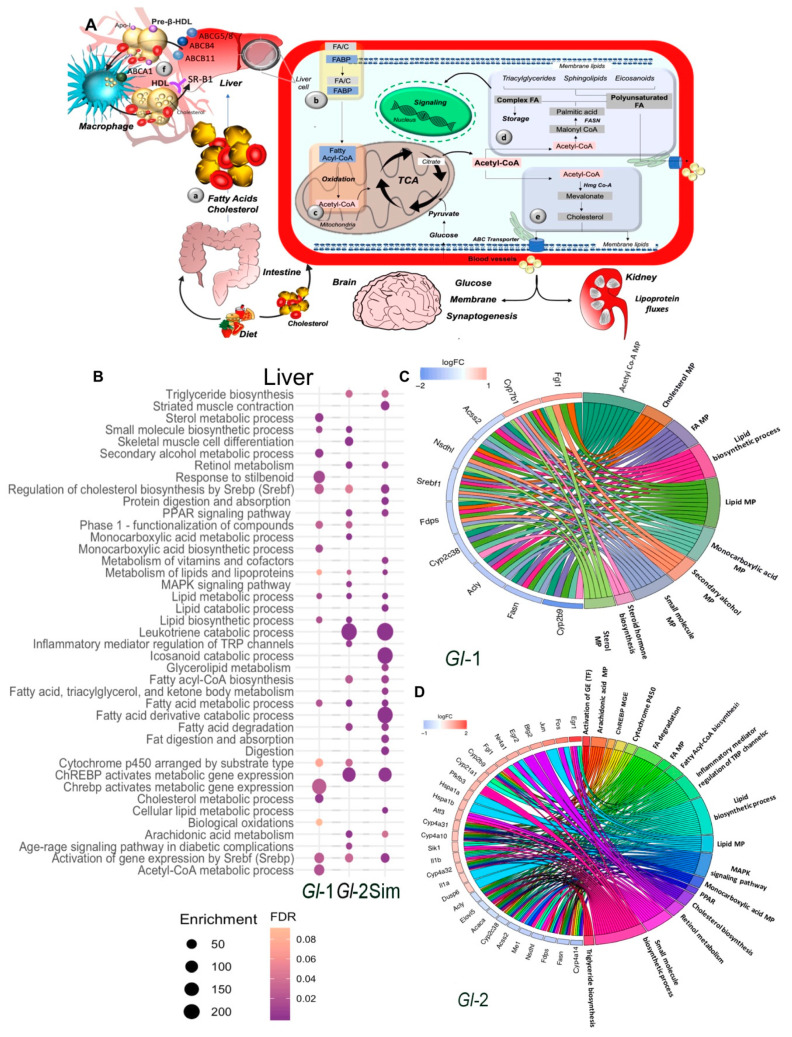
Main pathways of lipid metabolism altered by a high-cholesterol diet (HCD) given to C57BL/6 mice, including the administration of standardized hydroalcoholic extracts (*Gl*-1, *Gl*-2) from a Mexican strain of *Ganoderma lucidum*, in comparison with the drug simvastatin. (**A**) Schematic overview of major pathways involved in lipid metabolism. (a) Fatty acids (FAs) from dietary ingestion are either internalized in the cell from blood circulation by fatty acid-binding proteins (FABPs), or synthesized de novo within cells (b). (c,d) De novo lipogenesis begins with the conversion of acetyl-CoA to malonyl-CoA. The multifunctional enzyme fatty acid synthase (FASN) is then coupled to acetyl-CoA and malonyl-CoA to generate palmitic acid, which is further elongated and desaturated to produce diverse saturated and unsaturated FAs (d). These FAs can then be converted to diverse types of lipids, such as diacylglycerides and triacylglycerides via the glycerol phosphate pathway, as well as sphingolipids, phosphoinositides and eicosanoids, which have important signaling functions in cells and tissues. Another class of lipids are sterols, mainly cholesterol (e). (f) Macrophage reverse cholesterol transport. Liver and intestine synthesize apolipoprotein A-I (Apo-I) that load cholesterol in free pre-β-HDL molecules from peripheral tissues via macrophages ABCA1 transporter. These HDL complex particles mature and transport its cholesterol directly to the liver via the hepatic scavenger receptor class B type 1 (SR-B1) receptor. At systemic level, major organs involved in lipid metabolism are liver, skeletal muscle, and adipose tissue, which express specific receptors and transport proteins for lipids uptake. However, other organs are also involved, such as brain and kidney, which contribute in lipoprotein fluxes and lipid metabolism. TCA: tricarboxylic acid cycle. C: cholesterol. (**B**) Pathway enrichment analysis, represented in a bubble plot, of hepatic differentially expressed genes in mice liver fed with a HCD, treated with *Gl*-1 or *Gl*-2 extracts, or simvastatin. Bubble size represents the enrichment score, while their color represents false discovery rate (FDR) values. Circos plots showing the relationship between selected terms and annotated genes as indicated by connecting lines at *Gl*-1 (**C**) and *Gl*-2 (**D**) interventions. Genes are located on the left side of the graph, and ordered on the basis of their logFC values. MP: metabolic process.

**Figure 2 nutrients-13-00038-f002:**
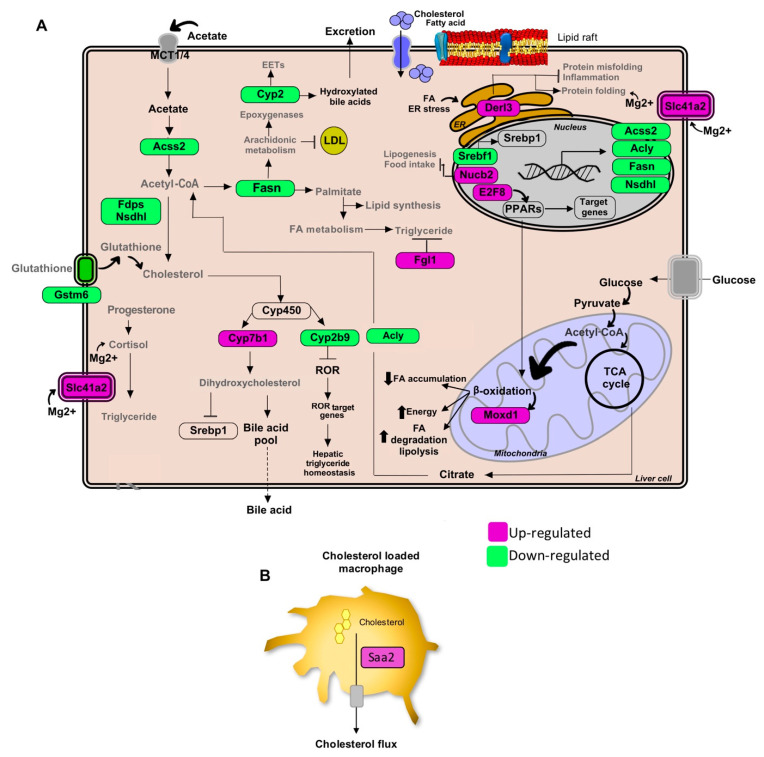
Pathways of lipid metabolism modulation in hepatic tissue by the *Gl*-1 extract of a Mexican strain of *Ganoderma lucidum*, under a high-cholesterol diet (HCD) given to C57BL/6 mice. (**A**) Lipid metabolic network and genes modulated by *Gl*-1 extract consumption. (**B**) Schematic outline of cholesterol efflux signaling in hepatic macrophages cells by the Saa2 gene. Green shaded boxes indicate significant down-modulation, while magenta shaded boxes indicate significant up-modulation of a gene in mice fed with a HCD and treated with *Gl*-1 extract versus mice fed with a HCD only. EETs: epoxyeicosatrienoic acids. ER: endoplasmic reticulum.

**Figure 3 nutrients-13-00038-f003:**
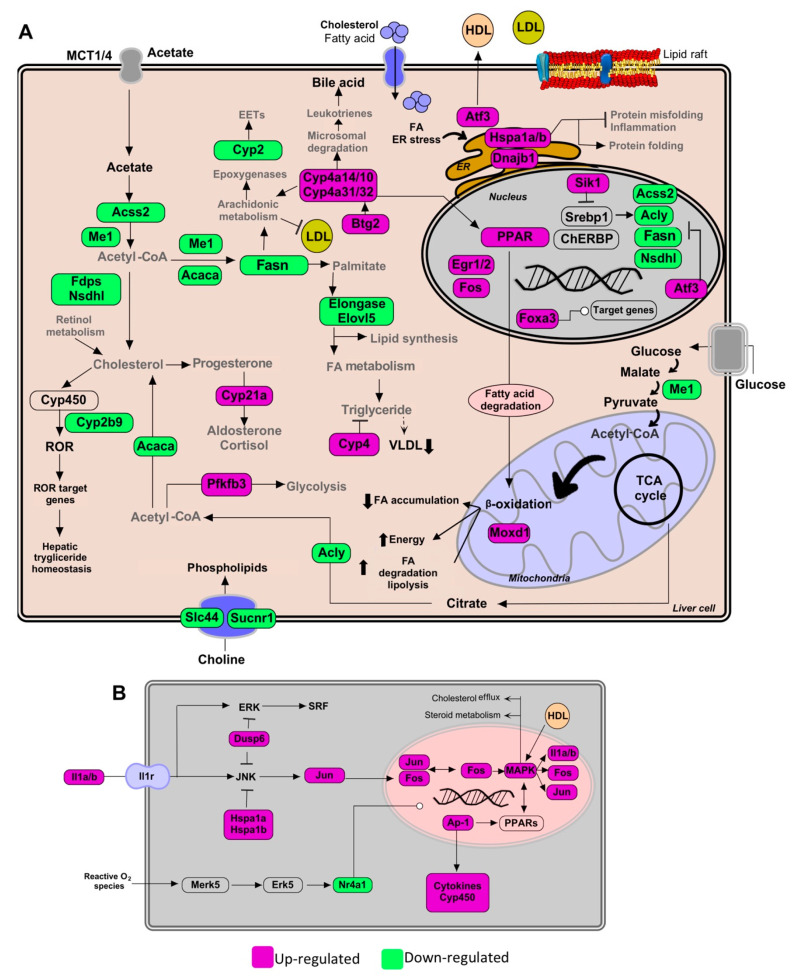
Pathways of lipid metabolism modulation in hepatic tissue by the *Gl*-2 extract of a Mexican strain of *Ganoderma lucidum*, under a high-cholesterol diet (HCD) given to C57BL/6 mice. (**A**) Lipid metabolic network and genes modulated by *Gl*-2 extract consumption. (**B**) Schematic outline of PPAR activation pathway up-modulated by higher expression levels of MAPK family genes and AP-1. Green shaded boxes indicate significant down-modulation, while magenta shaded boxes indicate significant up-modulation of a gene in mice fed with a HCD and treated with *Gl*-2 extract versus mice fed with a HCD only. EETs: epoxyeicosatrienoic acids. ER: endoplasmic reticulum.

**Figure 4 nutrients-13-00038-f004:**
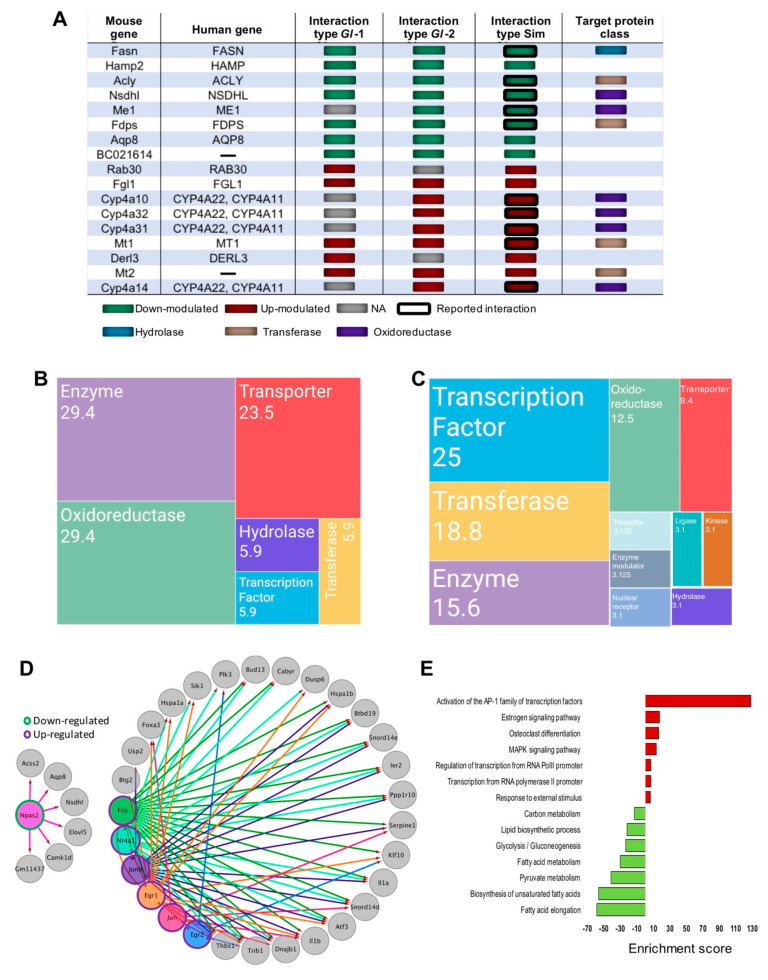
Genes associated to druggable metabolism, which are modulated by the intervention of *Ganoderma lucidum* extracts (*Gl*-1, *Gl*-2) or simvastatin (Sim), under a high-cholesterol diet (HCD) given to C57BL/6 mice, as well as the transcription factor (TF) networks established. (**A**) Druggable genes targeted by simvastatin as annotated by the DGIdb database, showing their type of interaction and target protein classes in comparison with those from *Gl* extracts. Green or magenta shaded boxes represent expression status in mice hepatic tissue or druggable genes. Mosaic plot showing the frequency of target classes modulated by *Gl*-1 (**B**) or *Gl*-2 (**C**) extract consumption. Each rectangle represents a category annotated by the Drug Repurposing Hub (www.broadinstitute.org/drug-repurposing-hub) and the DGIdb databases, while numbers within rectangles indicate percentage of druggable *Gl* modulated genes in each category. (**D**) Regulatory network of transcription factors and target genes. Shaded circles represent TFs, while gray circles indicate target genes. Outer green or magenta circles indicate the expression status of TFs. The regulatory network shows direct interactions between altered TFs and altered gene targets showing a significant positive correlation in their expression values. (**E**) Over-representation pathway analysis of annotated and correlated TF target genes. Bars show the fold-enrichment of the significantly enriched (*p* < 0.05) pathways. NA = Not available.

**Figure 5 nutrients-13-00038-f005:**
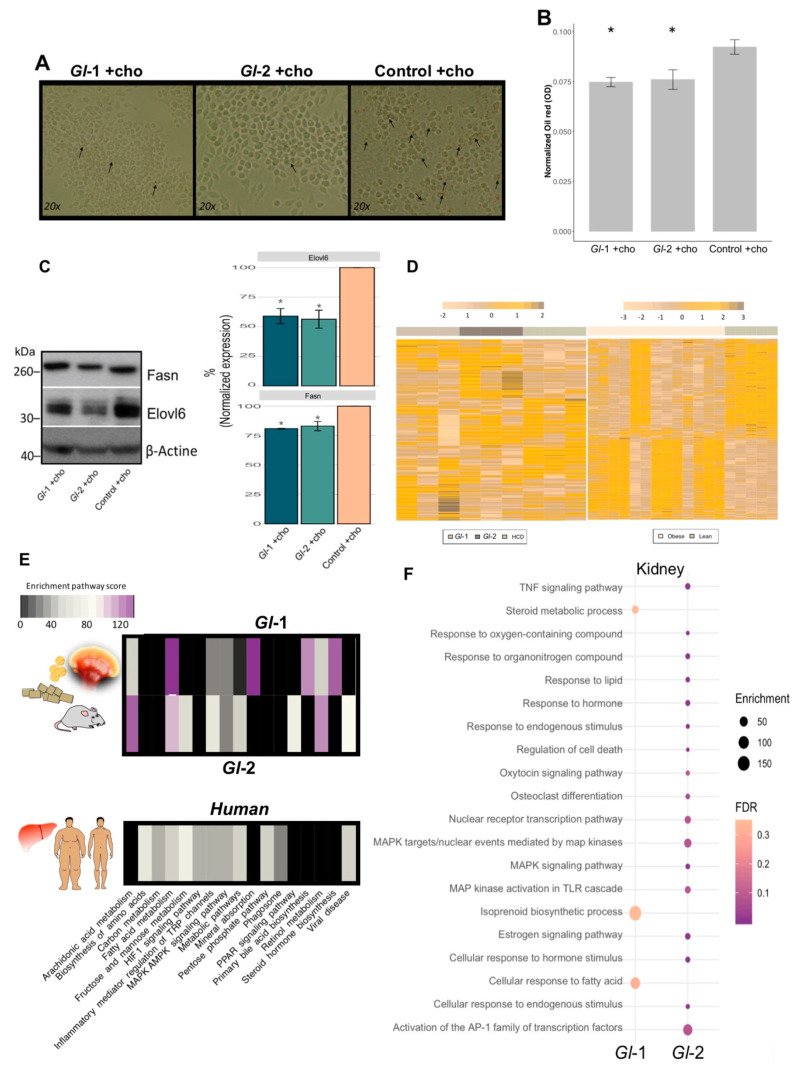
*Ganoderma lucidum* extracts (*Gl*-1, *Gl*-2) modulate the transcriptional landscape of hepatic-like macrophages and kidney tissue, as well as the common mouse-human metabolic process. (**A**) Oil red O staining of RAW 264.7 macrophages grown in high cholesterol conditions, after the intervention with *Gl*-*1* and *Gl*-*2* extracts, in comparison with the untreated control (vehicle). Zoom images obtained at 20x. Arrows indicate lipid accumulation. (**B**) Absorbance quantification of oil red O staining normalized by sulforhodamine B (SRB) absorbance levels of each experimental condition. Contrast: Absorbance of *Gl*-*1* or *Gl*-*2* extracts vs Control, * *p* < 0.05. (**C**) Western blot for Fasn and Elovl6 in RAW 264.7 macrophage-like cell line treated with *Gl* extracts (*Gl*-1, *Gl*-2), as well as the untreated control (vehicle), under high cholesterol concentrations. Representative blot from two independent experiments. Bar plot described the results of two independent experiments with three technical replicates represented as the mean +/− standard deviation error. * *p* < 0.05 (**D**) Comparative transcriptional landscape of homologous human and mouse genes modulated by both *Gl*-1 and *Gl*-2 extracts in liver tissue of C57BL/6 mice, under a high-cholesterol diet (HCD), in comparison with differentially expressed genes in liver tissue of obese versus lean patients. (**E**) Enriched pathways shared between hepatic genes modulated by *Gl* extracts (*Gl*-1, *Gl*-2) in mice, and altered hepatic genes in obese versus lean patients. (**F**) Bubble plot showing the pathway enrichment analysis of differentially expressed renal genes of C57BL/6 mice fed with a HCD, and treated with *Gl*-1 and *Gl*-2 extracts. The bubble size represents the enrichment score, while the bubble color indicates FDR values. Ch: cholesterol.

**Figure 6 nutrients-13-00038-f006:**
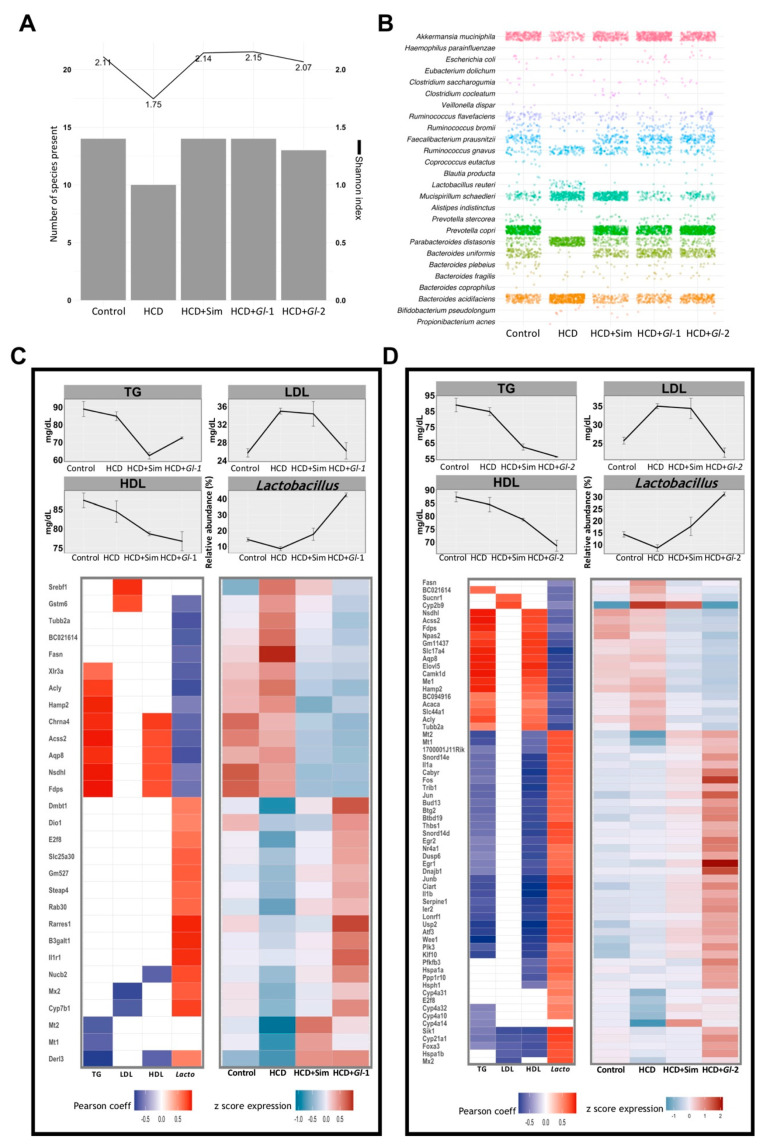
Hepatic genes modulated by *Ganoderma lucidum* extracts (*Gl*-1, *Gl*-2) correlated with biochemical parameters and relative abundance of Lactobacillus in the gut microbiome. (**A**) Gut bacterial richness at species level according to the Shannon index (line plot) and the number of detected species (bar plot) in each experimental group. (**B**) Frequency grid plot showing relative abundance of bacterial species in the gut microbiota grouped by experimental condition. (**C**,**D**) Pearson correlation between expression levels of genes significantly modulated by *Gl*-1 or *Gl*-2 extracts (lower panel) and blood levels (mg/dL) of TG, LDL, HDL, and the relative abundance (%) of Lactobacillus in gut microbiome composition (upper panel). The left heatmap shows significant correlations (Pearson coefficient > 0.5, in red; <−0.5, in blue; FDR < 0.1) between *Gl*-1 (**C**) or *Gl*-2 (**D**) responsive genes and their variables at the top. The right heatmap in *Gl*-1 (**C**) or *Gl*-2 (**D**) represents expression levels of correlated genes in each experimental condition (low to high expression scale is indicated in blue and red, respectively). TG: triglycerides. HDL: high-density lipoprotein cholesterol. LDL: low-density lipoprotein cholesterol. HCD: high-cholesterol diet. HCD + Sim = high-cholesterol diet + simvastatin. HCD + *Gl*-1 = high-cholesterol diet + *Gl*-1 extract. HCD + *Gl*-2 = high-cholesterol diet + *Gl*-2 extract.

## Data Availability

The data presented in this study are openly available in GEO Omnibus databse, reference number [GSE159656].

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
