# Peer review of "Mexican Ganoderma Lucidum Extracts Decrease Lipogenesis Modulating Transcriptional Metabolic Networks and Gut Microbiota in C57BL/6 Mice Fed with a High-Cholesterol Diet"

_nutrients, 2020, doi:10.3390/nu13010038_

Round 1
Reviewer 1 Report
The work presents interesting results of research from the field of medicine and nutrition. Medicine has recently recognized the potential of natural products in medicinal of people. Prevention of hyperlipidemia and associated diseases is a health priority in present time. High cholesterol concentration is a major risk factor for metabolic and cardiovascular diseases. Major public health problems (coronary heart diseases, obesity and metabolic syndromes) positively correlate with hypercholesterolemia. Thus, the prevention of lipid metabolism alterations is a very important topic. However, the work before printing, should be completed.
Introduction
In my opinion there is a lack of a clearly formulated aim of the work. Please completed it.
Methods
The description of statistical surveys is very general. Please provide the information what computer program was used, what tests?
Results
This section needs to be improved. It should include the authors' own results and their discussion. The results of other authors should be moved to the Discussion section.
Line 207-209: “Under normal physiological conditions, fatty acid (FA) metabolism involves a fine-tuned balance between anabolic and catabolic signaling that results in the production of important molecules and substrates for cellular function and homeostasis (Figure 1A) [21-26].” It is not clear whether this information regards to the research carried out in this work or is it data from the literature [21-26]?
Discussion
Figure 6. Does this Figure present the research carried out by other authors in discussion?
Author Response
Reviewer 1
The work presents interesting results of research from the field of medicine and nutrition. Medicine has recently recognized the potential of natural products in medicinal of people. Prevention of hyperlipidemia and associated diseases is a health priority in present time. High cholesterol concentration is a major risk factor for metabolic and cardiovascular diseases. Major public health problems (coronary heart diseases, obesity and metabolic syndromes) positively correlate with hypercholesterolemia. Thus, the prevention of lipid metabolism alterations is a very important topic. However, the work before printing, should be completed.
We would like to thank the reviewer for his/her comments and useful observations.
Introduction
In my opinion there is a lack of a clearly formulated aim of the work. Please completed it.
We appreciate the reviewer´s observation. We have, accordingly fixed our introduction to state in clearer way the objective and what we consider to be the main novelty of our work.
Specifically, in line 99, now reads as:
“A more complete molecular characterization is needed to understand the underlying mechanisms of action of Gl on lipid metabolism, particularly for Mexican genetic resources under differing cultivation conditions. In this study, we performed whole gene expression analysis of liver and kidney tissues of C57BL/6 mice fed with a high-cholesterol diet (HCD) and the administration of standardized extracts from a Mexican Gl grown on a substrate with and without ASA. Then we explored correlations between RNA profiles established by the consumption of Gl extracts and biochemical features, as well as microbiota composition, aiming at providing a comprehensive view of molecular mechanisms linked to Gl hypocholesterolemic effects. On the basis of our results, we present a highly detailed landscape, previously unrecognized, of transcriptional changes occurring in liver and kidney associated to the administration of Gl extracts. In both Gl extracts, we observed a positive enrichment in molecular programs relevant for the degradation and excretion of fatty acids. Importantly, we show that the addition of ASA to the Gl cultivation substrate increases transcriptional responses that recapitulate those exerted by the simvastatin drug and are enriched to a higher degree in processes and pathways related to human obesity alterations in liver. We also revealed a profile of Gl-modulated genes that is associated with Lactobacillus enrichment and the reduction of cholesterol-derived fats in blood mediated by the Gl interventions. This study provides new evidence that adds and extends the knowledge about the effects of natural resources, such as Gl, over transcriptional changes and the role of substrate composition for maximizing beneficial properties to prevent metabolic disorders associated to hypercholesterolemia”
Methods
The description of statistical surveys is very general. Please provide the information what computer program was used, what tests?
As requested, this section has been amended with the incorporation of more details on this aspect. Other changes along the methods section has been made with the intention of improve the clarity of our statistical and computational analysis.
In particular, the paragraph referring to the statistical analysis in line 294, reads as:
2.19 Statistical analysis
“Statistical differences across the experimental conditions were computed by a non-parametric analysis using Wilcoxon or Kruskal-Wallis test, to compare two groups or multiple groups, respectively, using R software. A p-value ≤ 5% was considered significant. P-value: * < = 0.05, ** < = 0.01, *** < = 0.001, **** < = 0.0001. For microarray analysis, a moderated t-test was performed and false discovery rate (FDR) was applied to adjusted p values for multiple hypothesis on limma library of Bioconductor/R environment.”
Results
This section needs to be improved. It should include the authors' own results and their discussion. The results of other authors should be moved to the Discussion section.
Thank you for your comment. In accordance with your suggestion, we have re-structured the Results sections. We have now, to the best of our effort, revised and removed paragraphs and sentences, which appeared to be discussion on another author´s reports. These subsections have been instead, included within Discussion section.
Line 207-209: “Under normal physiological conditions, fatty acid (FA) metabolism involves a fine-tuned balance between anabolic and catabolic signaling that results in the production of important molecules and substrates for cellular function and homeostasis (Figure 1A) [21-26].” It is not clear whether this information regards to the research carried out in this work or is it data from the literature [21-26]?
The described text is a general description of lipid metabolism in mammal cells based on what is reported in the literature that we retained important to present in order to introduce our own results within this metabolic pathways, both between HCD diet vs normal diet, and Gl-1, Gl-2 and Simvastatin consumption vs HCD condition. We maintain that this information is useful for the reader to get a more comprehensive view of the described phenotype.
Discussion
Figure 6. Does this Figure present the research carried out by other authors in discussion?
No, as state in the text the result of plots in Figure 6 were entirely constructed using data retrieved from experiments carried out by our research group and described within the corresponding subsections (i.e.,2.15 the Gut microbiota analysis in line 253, 2.18 Blood systemic parameters, in line 289 and 2.5 Gene expression profiles, in line 166)
Reviewer 2 Report
The article by Sandra L. Romero-Córdoba and colleagues investigates how transcriptional metabolic networks and gut microbiota are involved in Ganoderma lucidum extracts against metabolic disorders both in vivo and in vitro. This work has a potential interest and fits with the scope of the journal. However, the following issues should be clarified further.
Major points:
- The finding of Ganoderma lucidum in metabolic disorders, especially in terms of gut microbiota and metabolic networks, is not new, which has been reported by many other investigations. The authors should emphasize potential novel findings of this work to convince the scientific community and potential readers outside the field about the novelty of the paper.
- The authors should improve the abstract, particularly the crucial outcomes should be emphasized.
- What’s the major biochemical composition of standardized Gl extracts? What kind of method is applied to detect biochemical composition. It should be clearly and simply stated in this manuscript.
- For the benefit of the reader, the function and background of acetylsalicylic acid added to Gl-2 should be clarified.
- The rationale of dose application of Gl-1 extract, Gl-2 extract, and simvastatin directly on C57BL/6 mice in animal treatment is not clear.
- In western blotting analysis (Figure C), how can the author get the results of 81%, 80%, 54%, and 61%? Still, one single experiment is not enough. For the statistical reason, generally, at least three repetitions are required for western blotting.
Minor points:
- In each experiment, more details should be included. For example, line 118: how much RNA is used in the gene expression profiles; line 172: how much proteins are used in WB.
- the full name should be given out at the first mention, such as Line 142: TF.
- Line 159: CO2 instead of CO2.
- Figure 5C: What does it mean, 40, 30, 260?
- Line 418: *p<0.05 versus?
- The subtitle format in each section is not consistent with the journal of nutrients.
- Characters in Figure 1B and Figure 5F are too small to read.
Author Response
Reviewer 2
The article by Sandra L. Romero-Córdoba and colleagues investigates how transcriptional metabolic networks and gut microbiota are involved in Ganoderma lucidum extracts against metabolic disorders both in vivo and in vitro. This work has a potential interest and fits with the scope of the journal. However, the following issues should be clarified further.
First for all, we take the opportunity to thank the reviewer for his/her comments and useful observations.
Major points:
The finding of Ganoderma lucidum in metabolic disorders, especially in terms of gut microbiota and metabolic networks, is not new, which has been reported by many other investigations. The authors should emphasize potential novel findings of this work to convince the scientific community and potential readers outside the field about the novelty of the paper.
We appreciate this comment. In order to substantiate the scientific value of the present manuscript, we have undertaken some modifications and additions, specially at the introduction. In particular, we have included the next paragraphs.
Paragraphs in line 63:
“In fact, the role of Gl extracts as modulators of cholesterol metabolism has been recognized and already investigated in clinical trials [9]. In addition, research works have studied the modulation of genes and protein expression exerted by Gl [10,11], while others have focused on changes in microbiota composition [8,12]. Most reports have centered their attention on evaluating the expression of only few pre-selected genes, while only a few have studied global expression changes in the liver to understand the effects of Gl for counteracting hypoglycemia [13] or neoplastic development [14]. However, none of them has addressed global transcriptomic changes and their association with physiological, cellular and gut microbiome conditions related to the hypocholesterolemic effects of Gl.We believe this paragraph serves as preamble to our work, highlighting the state of the current knowledge about molecular mechanism induced by Gl, which undelay its properties at counteracting hyperlipidemia.”
And In line 99:
“A more complete molecular characterization is needed to understand the underlying mechanisms of action of Gl on lipid metabolism, particularly for Mexican genetic resources under differing cultivation conditions. In this study, we performed whole gene expression analysis of liver and kidney tissues of C57BL/6 mice fed with a high-cholesterol diet (HCD) and the administration of standardized extracts from a Mexican Gl grown on a substrate with and without ASA. Then we explored correlations between RNA profiles established by the consumption of Gl extracts and biochemical features, as well as microbiota composition, aiming at providing a comprehensive view of molecular mechanisms linked to Gl hypocholesterolemic effects. On the basis of our results, we present a highly detailed landscape, previously unrecognized, of transcriptional changes occurring in liver and kidney associated to the administration of Gl extracts. In both Gl extracts, we observed a positive enrichment in molecular programs relevant for the degradation and excretion of fatty acids. Importantly, we show that the addition of ASA to the Gl cultivation substrate increases transcriptional responses that recapitulate those exerted by the simvastatin drug and are enriched to a higher degree in processes and pathways related to human obesity alterations in liver. We also revealed a profile of Gl-modulated genes that is associated with Lactobacillus enrichment and the reduction of cholesterol-derived fats in blood mediated by the Gl interventions. This study provides new evidence that adds and extends the knowledge about the effects of natural resources, such as Gl, over transcriptional changes and the role of substrate composition for maximizing beneficial properties to prevent metabolic disorders associated to hypercholesterolemia”
In which we sought to emphasize our approach and scientific contribution
Additionally, we have included the next two paragraphs within our Discussion.
One, in line 592:
“Accumulating evidence from previous biological and pharmacological research demonstrated that Gl extracts and their bioactive compounds show anti-hypercholesterolemic, hypoglycemic [6], and antioxidant activities [7], and modulate the gut microbiota [8]. However, most studies only evaluate a limited set of genes, and it remains unknown how G. lucidum modulates whole cell transcriptional-programs. To the best of our knowledge, this study presents the first comprehensive transcriptional landscape of Gl-modulated genes in hepatic and renal tissue of a murine model under HCD, which allows us to propose how standardized extracts from Mexican Gl operate on and identify unique molecular mechanisms, treatment-targets, similarities to the lowering-cholesterol Simvastatin drug mechanisms, and correlations with relevant circulating biochemical parameters and gut microbiota composition, including differences across Gl extracts obtained from basidiocarps cultivated on different substrates.”
And in line702:
“Potential sources of variability in product chemistry and efficacies need to be addressed in the natural products manufacturing [61]. In this context, we confront this challenge by studying well characterized patented Gl extracts, and confirming the molecular identification of Mexican Ganoderma lucidum. This has been a relevant bias in other studies that analyzed commercial extracts labeled as Ganoderma species that are not correctly assigned [61]. We characterized the strain studied of Ganoderma lucidum by sequencing the ITS1-5.8S-ITS2 region from the rDNA, showing that it clearly separated in the consensus phylogenetic trees of reference strains from European, North American and Southeast Asian origin [15]. Our work therefore shows new data from an independent genotype on how cellular expression portraits of hepatic and renal tissues are modulated in response to Gl extracts treatment and HCD diet. We recognized well-known driver Gl-target genes, as well as novel contributing transcriptional circuits not well previously described due to limited high-throughput information”
- The authors should improve the abstract, particularly the crucial outcomes should be emphasized.
We have likewise taken the reviewer advise and modified our abstract in a way we consider highlight our main findings and contributions. It now reads as follows:
“Prevention of hyperlipidemia and associated diseases is a health priority. Natural products, such as the medicinal mushroom Ganoderma lucidum (Gl), have demonstrated hypocholesterolemic, prebiotic and antidiabetic properties. However, the underlying transcriptomic mechanisms by which Gl exerts bioactivities are not completely understood. We report a comprehensive hepatic and renal transcriptome profiling of C57BL/6 mice under the consumption of a high-cholesterol diet and two standardized Gl extracts obtained from basidiocarps cultivated on conventional substrate (Gl-1) or substrate containing acetylsalicylic acid (ASA; Gl-2). We showed that Gl extracts modulate relevant metabolic pathways involving the restriction of lipid biosynthesis and the enrichment of lipid degradation and secretion. The Gl-2 extract exerts a major modulation over gene expression programs showing the highest similarity with simvastatin druggable-target-genes and are enriched higher in processes related to human obesity alterations in liver. We further show a subset of Gl-modulated genes correlated with Lactobacillus enrichment and the reduction of circulating cholesterol-derived fats. Moreover, Gl extracts induce a significant decrease of macrophage lipid storage, which occurs concomitantly with the down-modulation of Fasn and Elovl6. Collectively, these evidences suggest a new link between Gl hypocholesterolemic and prebiotic activity, revealing thereby that standardized Mexican Gl extracts are a novel transcriptome modulator to prevent metabolic disorders associated to hypercholesterolemia.”
- What’s the major biochemical composition of standardized Gl extracts? What kind of method is applied to detect biochemical composition. It should be clearly and simply stated in this manuscript.
The complete biochemical composition of Gl standardized extract, as well as the analytical methods employed to determined it, was reported within supplementary table 1 (ST1.xlsx) in the original version. With the purpose of indicating in a clearer way this information within our manuscript, we have included a dedicated paragraph within Methods section, in line 131, that reads as:
“2.2 Biochemical composition of standardized Gl extracts
Biochemical composition of standardized Gl extracts, as well as a the analytical methods and techniques performed to estimate each nutrient and organic profile are described in Table S1”
3.For the benefit of the reader, the function and background of acetylsalicylic acid added to Gl-2 should be clarified.
Thank you for this observation. To properly address this concern, we have included a paragraph in the Introduction with the purpose of providing more information on the rationale behind the culture supplementation with ASA.
The aforementioned paragraph has been included in line 90 in introduction section, which reads:
“A novel approach is that one of our Gl extracts derives from basidiocarps cultivated in substrate containing acetylsalicylic acid (ASA). It has been shown that this drug exerts multiple effects on plant development and is capable of promoting the synthesis of secondary metabolites [17]. The effect of ASA on the functional and medicinal properties of Gl extracts has not yet been studied at genomic level. In similar studies, the cultivation of Agaricus bisporus with ASA prevents post-harvest browning by maintaining the integrity of the membrane and facilitates the accumulation of phenols [18]. ASA supplementation on Gl enhances the levels of polysaccharides, triterpenes and ganoderic acid [19-21].
And 650 line in the discussion section
“There are no previous studies on the transcriptional effect of Gl extracts obtained from basidiocarps cultivated on substrate containing ASA. Our data demonstrated that specific mechanisms exist for Gl-2 extracts, such as improved modulation of gene-expression programs, probably established by a fine tuning of core transactional factor networks that lead to a more complex transcriptional effect. Even more, we detected more robust similarities between Gl-2 and Simvastatin gene-expression activity. Our recent study demonstrated that the addition of ASA in the Gl cultivation substrate stimulates the production of more or new bioactive compounds, mainly the polysaccharides, B-methyl glucopyranoside, DL-arabinitol and ribitol, in comparison to the conventional extract (no ASA) [50]. This pattern might be the result of biotransformation processes that occur when fungal enzymes catalyze chemical modifications of a compound [51]. From a biomedical point of view, we focused on the acetylsalicylic acid, a non-steroidal anti-inflammatory drug approved for the treatment of pain and fever, that has also been included as a secondary prevention in stroke treatment and cardiovascular disease at low dose. ASA has also demonstrated improved effects when its administrated in combination with other cholesterol-lowering drugs, such as Simvastatin. Coadministration of both drugs reduces the incidence of cardiovascular disease and improves blood circulation [52,53]. A proposed mean to get this synergistic effect is probably by an improved solubility of Simvastatin by ASA that consequently enhances its bioavailability [54]. How ASA was biotransformed by Gl or how ASA non-converted compounds might maximize the Gl-2 extract need to be further addressed.”
- The rationale of dose application of Gl-1 extract, Gl-2 extract, and simvastatin directly on C57BL/6 mice in animal treatment is not clear.
The rationale of the experimental design, as well as diet composition, preparation, and administration, are now described. Two paragraphs and experimental details were added in this section, for example:
“Hypocholesterolemic and prebiotic effects of standardized Gl extracts were assessed in mice fed with a high-cholesterol diet, in comparison with the simvastatin drug. The effect of ASA, added to the substrate used for mushroom cultivation, on the functional and medicinal properties of Gl extracts was also studied (Gl-1: no ASA; Gl-2: 10 mM ASA). The concentrations of cholesterol (0.5%), simvastatin (0.03 g/100 g), as well as lyophilized Gl-1 and Gl-2 extracts (1.0%), in the animal diet were previously determined by the authors [10]…”
“…Concentrations studied of cholesterol, simvastatin or Gl extracts were added to the food pellet and mixed homogeneously, depending on each experimental treatment. Animals were fed ad libitum with AIN-93 standard and experimental diets (average intake: 3.47±0.22 g/day). Water was also provided ad libitum…”
- In western blotting analysis (Figure C), how can the author get the results of 81%, 80%, 54%, and 61%? Still, one single experiment is not enough. For the statistical reason, generally, at least three repetitions are required for western blotting.
We agree with the notion that a single experiment is insufficient. In fact, this information was erroneously omitted in our first draft, since two different WB experiments were performed with 3 technical replicates each. We have appended a new barplot accompanying the blot results. The captions referring to this figure now reads as:
“Representative blot from two independent experiments. Barplot described the results of two independent experiments with three technical replicates represented as the mean +/− standard deviation error. * p<0.05”
Minor points:
- In each experiment, more details should be included. For example, line 118: how much RNA is used in the gene expression profiles; line 172: how much proteins are used in WB.
In accordance with your observation, we have fixed the corresponding section, to include this missing information regarding the details of our experiments.
- the full name should be given out at the first mention, such as Line 142: TF.
The line has been modified accordingly. Likewise, we have taken great care at trying to detect and change similar insertions along the manuscript
- Line 159: CO2 instead of CO2.
The line has been modified accordingly
- Figure 5C: What does it mean, 40, 30, 260?
The scale in this figure refers to molecular weight (KDa), which is now clearly indicated within the figure
- Line 418: *p<0.05 versus?
This line has been amended. It now reads as: “p <0.05 vs Control”
- The subtitle format in each section is not consistent with the journal of nutrients.
Thank you again, we have reformatted our manuscript to comply with the journals format
- Characters in Figure 1B and Figure 5F are too small to read.
We have increased the size of the characters, as much as possible, given the limitation of space
Round 2
Reviewer 1 Report
The manuscript is carefully prepared and is suitable for publicationin its current form.